# The Influence of Ag$^+$/Ti$^{4+}$ Ratio on Structural, Optical and Photocatalytic Properties of MWCNT–TiO$_2$–Ag Nanocomposites

Ramona-Crina Suciu, Mioara Zagrai, Adriana Popa, Dana Toloman, Camelia Berghian-Grosan, Cristian Tudoran and Maria Stefan *

National Institute for Research and Development of Isotopic and Molecular Technologies, Donat Street, No. 67–103, 400293 Cluj-Napoca, Romania; ramona.suciu@itim-cj.ro (R.-C.S.); mioara.zagrai@itim-cj.ro (M.Z.); adriana.popa@itim-cj.ro (A.P.); dana.toloman@itim-cj.ro (D.T.); camelia.grosan@itim-cj.ro (C.B.-G.); cristian.tudoran@itim-cj.ro (C.T.)
* Correspondence: maria.stefan@itim-cj.ro

**Abstract:** In this paper, we propose a simple procedure to obtain multi-walled carbon nanotubes (MWCNTs) decorated with TiO$_2$–Ag nanoparticles (MWCNT–TiO$_2$–Ag). The MWCNTs were decorated with TiO$_2$–Ag via combined functionalization with –OH and –COOH groups and a polymer-wrapping technique using poly(allylamine)hydrochloride (PAH). TiO$_2$-modified Ag nanoparticles were synthesized via the Pechini method using a mixture of acetylacetonate-modified titanium (IV) isopropoxide with silver nitrate (with Ag$^+$/Ti$^{4+}$ atomic ratios of 0.5%, 1.0%, 1.5%, 2.0%, and 2.5%) and L(+)-ascorbic acid as reducing agents. XRD analysis revealed the formation of nanocomposites containing CNT, anatase TiO$_2$, and Ag. The presence of nanoparticles on the MWCNT surfaces was determined using TEM. The morphology of the TiO$_2$–Ag nanoparticles on the MWCNT surfaces was also determined using TEM. UV–Vis investigations revealed that an increase in the ratio between Ag$^+$ and Ti$^{4+}$ decreased the band gap energy of the samples. The characteristic vibrations of the TiO$_2$, Ag, and C atoms of the graphite were identified using Raman spectroscopy. The photocatalytic activity of the MWCNT–TiO$_2$–Ag nanocomposite was assessed by examining the degradation of Allura Red (E129) aqueous solution under UV irradiation. The dye photodegradation process followed a pseudo-first-order kinetic with respect to the Langmuir–Hinshelwood reaction mechanism. The spin-trapping technique evidenced that $\bullet$O$^{2-}$ was the main species generated responsible for the Allura Red degradation.

**Keywords:** Ag-doped TiO$_2$; MWCNT; Allura Red





## 1. Introduction

Semiconductor photocatalysis is a cost-effective and eco-friendly de-pollution technology for removing different organic pollutants in wastewater [1]. Titanium dioxide (TiO$_2$) is among the most promising photocatalyst due to its intrinsic properties, high chemical and thermal stability, biocompatibility, and low cost, and it is widely used in many industrial applications, such as environmental purification, the production of pigments, catalyst support coatings, solar cells, and biomaterials or the decomposition of carbonic acid gas [2–8]. However, TiO$_2$ possesses disadvantageous qualities, such as its specificity to UV light and rapid recombination of charge carriers produced under irradiation with light of specific wavelengths [9,10]. Thus, different strategies have been developed to improve the electron–hole separation of TiO$_2$ and prolong its absorption of visible light. Depositing noble metals, such as Ag, Au, Pt, and Pd, on the surface of TiO$_2$ could enhance its photocatalytic efficiency because they act as electron traps, promoting interfacial charge transfer processes in the composite systems [11–14].

Additionally, the excellent electronic properties of MWCNTs provide a continuous electronic state in the conduction band, ideal for transferring electrons. This facilitates the migration of excited electrons into the MWCNTs, thereby increasing their photocatalytic activity under visible light [15]. Because of the unique properties of composite materials resulting from combining two entities with different functions, many systems based on MWCNTs and semiconductors have been studied [16–18].

Due to rapid industrialization, numerous pollutants, such as dyes, antibiotics, and pesticides, are released into surface water, with devastating consequences for the water environment. For example, many dyes have carcinogenic and mutagenic effects, which can damage aquatic organisms [19–21].

A review of the literature highlights several nanostructured materials that can be used for the degradation or detection of azo dyes, including sunset yellow [22], Allura Red [23], Ponceau 4R, and Carmoisine [24,25]. Allura Red is a popular red azodye used worldwide as a food dye in many products, such as cotton candy, soft drinks, cherry-flavoured products, and children's medication [7]. Recently, there have been an increasing number of studies on the safety of these types of dyes, and it has been discovered that they can be hazardous to human health and aquatic environments [26]. Considerable research has been conducted on the genotoxic properties of these dyes, and the results remain controversial [26]. In addition, many depollution and detection strategies have been developed to solve the environmental problems caused by dye-related industrial effluents [27,28].

Few studies appear to have been conducted on nanocomposites containing MWCNTs coated with $TiO_2$–Ag, and the sol-gel method, with different precursors and reagents, is the most used synthesis method [29,30]. These nanocomposites had a strong antibacterial activity toward E. coli and S. aureus. Ag–MWCNT–$TiO_2$ core–shell ternary nanocomposites were successfully synthesized using a facile one-pot synthesis. The effect of the Ag content on the photoreactivity of the core–shell samples in the photoreduction of $CO_2$ was evaluated [31]. Other studies have revealed the photocatalytic activity of an Ag/$TiO_2$/MWCNT ternary nanocomposite toward methylene blue degradation [31].

In the context outlined above, the originality of our proposed approach consists in obtaining MWCNT–$TiO_2$–Ag nanocomposites through the combination of two facile synthesis methods. The Pechini method is used to obtain TiO2–Ag nanoparticles, which are subsequently used to decorate functionalized MWCNTs. Specific functionalization with -OH and –COOH groups using poly(allylamine)hydrochloride (PAH) as the polymer binder allows us to obtain stable and well-dispersed MWCNT–$TiO_2$–Ag nanocomposites. Specific experimental conditions were used, and a complex investigation was performed to determine the properties of the new nanocomposite materials, e.g., to improve photocatalytic efficiency. The photocatalytic degradation of Allura Red (E129) was investigated under visible light irradiation. The photocatalytic mechanism was established based on reactive oxygen species (ROS) and scavenger experiments.

## 2. Results

### 2.1. Investigation of Precursors

Thermal analyses of dried precursors were performed to establish the optimum crystallization temperature for the corresponding $TiO_2$ oxides. Figure 1 shows the thermogravimetric (TG), differential thermal analysis (DTA), and derivative thermogravimetric (DTG) curves of the dried (at room temperature for 10 days) titanium precursor solution in a temperature range of 20 °C–1200 °C and at a heating rate of 10 °C/min in an airflow.

Because the quantity of Ag doping does not significantly influence the thermal transformations and total mass losses of the samples, we chose the sample with the highest percentage of Ag in its $TiO_2$ matrix ($Ag^+/Ti^{4+}$ = 2.5%) for the thermal investigation.

The thermal decomposition of $TiO_2$ precursors takes place in different progressive steps of temperature (TG) depending on physical and chemical processes that occur with increasing temperature. The total weight loss (44.75%) associated with the transformation of precursors into $Ag^+/Ti^{4+}$ = 2.5% is the result of physical and chemical adsorbed water

(8.07%), volatilization and combustion of residual water and organic parts from polymeric gels (20.63% gr), combustions of residual carbon (6.45%), Ag-crystallization of amorphous anatase (9.6%). The corresponding thermal effects are shown in the DTA curve. Thus, significant effects corresponding to the exothermic peak at ~420.43 °C in the DTA curve are related to the beginning of crystallization of $TiO_2$ anatase, whereas that around 543.57 °C indicates the phase transformation anatase–rutile. These transformations are confirmed by the transformations corresponding to the DTG curves.

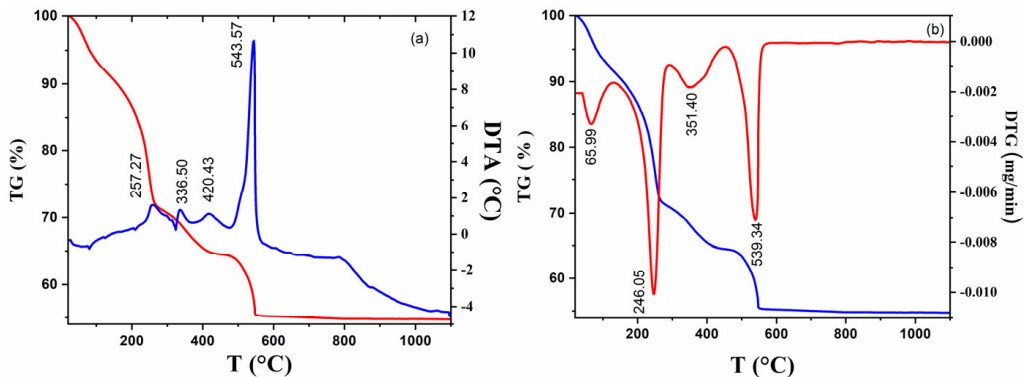

**Figure 1.** (**a**) Thermogravimetric (TG), differential thermal analysis (DTA), and (**b**) derivative thermogravimetric (DTG) curves of the dried (100 °C) titania precursors ($Ag^+/Ti^{4+}$ = 2.5%).

To further investigate the $TiO_2$ precursors, the FT–IR spectra of the precursor solution containing $Ag^+/Ti^{4+}$ = 2.5% are presented in Figure 2. The absorption shoulders at 1634 cm$^{-1}$ are attributed to bending vibrations of the -OH groups, and the bands centered at 1580, 1530, 1430, and 1025 cm$^{-1}$ corresponded to conjugated C-O vibrations [C=C-(C=O)] and [C=C-(C-)O-] of the acac ligand. The peak at 1355 and 931 cm$^{-1}$ is related to the $CH_2$ symmetric vibration and $CH_3$-C-$CH_3$ stretching modes of the isopropoxy group. The band at 1126 cm$^{-1}$ is due to Ti-O-C stretching vibration, the broad peaks in the range 653–550 cm$^{-1}$ and 495–436 cm$^{-1}$ indicated the presence of $\nu_{Ti-O}$ and $\nu_{Ti-O-Ti}$ bond, respectively, and the peak located at 443 cm$^{-1}$ is correlated with the asymmetric stretching vibration of Ti-O-Ag [32].

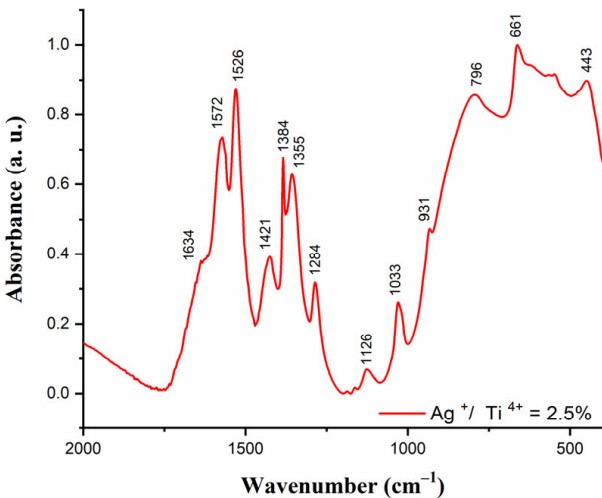

**Figure 2.** The FT–IR spectra of precursor solution containing $Ag^+/Ti^{4+}$ = 2.5%.

### 2.2. Structural, Morphologic, and Optical Properties

The crystalline structure and structural parameters of MWCNT–$TiO_2$–Ag nanocomposites were confirmed by XRD, as shown in Figure 3a. The indexed diffraction peaks

correspond to the tetragonal anatase crystalline phase (PDF card 01-089-4921) and Ag (PDF card 00-002-1098). No other peaks related to secondary phases were observed in the diffractograms. Devi et al. [33] reported that Ag-deposited nitrogen-doped $TiO_2$ leads to the same effects. The peak at la $2\theta = 26.4°$ is typical for the 002 direction of graphite [34] (file PDF 01-075-0444) and is probably superposed with (101) at 25.38° of anatase [35,36]. The small right—shifting to higher $2\theta$ values of diffraction peaks of anatase (101) with the increase in Ag content (Figure 3b) less than 0.1° was reported by Yuliati et al. [37] and was assigned to the presence of Ag on $TiO_2$ with minimal crystal distortion. The elemental chemical analysis also underlines the presence of MWCNT (Figure 3).

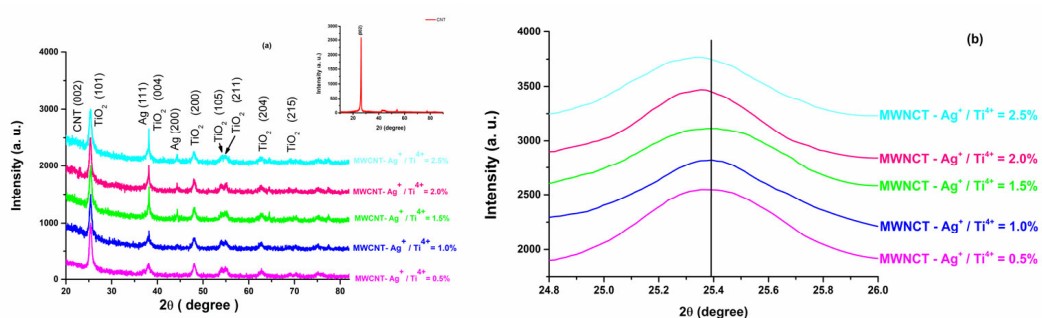

**Figure 3.** XRD diffraction patterns of MWCNT–$TiO_2$–Ag nanocomposites. (**a**) $2\theta = 20$–80° and (**b**) $2\theta = 24.8$–26°.

The Warren—Averbach X-ray profile Fourier analysis of the (101) ($2\theta = 28.31°$), (200) ($2\theta = 48.05$), and (215) ($2\theta = 75.058°$) anatase peak profiles were processed by the XR-LINE [38] computer program to determine the effective crystallite mean size ($D_{eff}$). The crystallite size distribution function was determined from the second derivative of the strain-corrected Fourier coefficients [39]. Also, from this profile, we obtained the root mean square (RMS) of the microstrains averaged along the real space distance, $\langle \varepsilon^2 \rangle_{hkl}^{1/2}$ [40].

The elementary parameters of the cell were determined by Rietveld type refinement using the Powder Cell program [41], developed by Werner Kraus and Gert Nolze (BAM Berlin).

Table 1 summarizes the microstructural parameters of $TiO_2$ anatase nanoparticles from the MWCNT–$TiO_2$–Ag.

**Table 1.** The effective, crystalline mean size, $D_{eff}$ (nm), the root mean square (RMS) of the microstrains averaged along the real space, $\langle \varepsilon^2 \rangle_{hkl}^{1/2}$.

| Sample | Unit Cell Parameter | | Cell Volume [Å³] | Effective Crystalline Mean Size, $D_{eff}$ (nm) | Microstrains Averaged along the Real Space, $\langle \varepsilon^2 \rangle_{hkl}^{1/2} \times 10^3$ |
|---|---|---|---|---|---|
| | a [Å] | c [Å] | | | |
| MWCNT-$Ag^+$/$Ti^{4+}$ = 0.5% | 3.7717 | 9.4603 | 134.576 | 14.21 | 11.3 |
| MWCNT-$Ag^+$/$Ti^{4+}$ = 1.0% | 3.7824 | 9.4821 | 135.656 | 14.08 | 12.44 |
| MWCNT-$Ag^+$/$Ti^{4+}$ = 1.5% | 3.7892 | 9.4888 | 136.240 | 13.86 | 13.86 |
| MWCNT-$Ag^+$/$Ti^{4+}$ = 2.0% | 3.7920 | 9.4842 | 136.375 | 13.35 | 14.2 |
| MWCNT-$Ag^+$/$Ti^{4+}$ = 2.5% | 3.7951 | 9.5046 | 136.892 | 13.18 | 15.6 |

Microstructural data shows that as the dopant concentration increases, the average crystallite decreases from 14.21 to 13.18 nm. The decrease in crystallite size is due to the effects of Ag at the boundary of $TiO_2$ [42].

## 2.3. FT–IR Spectroscopy

Figure 4 presents the FT–IR spectra obtained for MWCNT–$TiO_2$–Ag nanocomposites with various $Ag^+$/$Ti^{4+}$ ratios. In the FT–IR spectra, the peak from 3414 cm$^{-1}$ can be associated with the -OH stretching vibrations. The C-H stretching mode from the

MWCNT structure [43] is evidenced in all the MWCNT–TiO$_2$–Ag samples at about 2852 and 2924 cm$^{-1}$ (Figure 4). The bending mode of the adsorbed water molecules on the nanocomposites is responsible for the 1637 cm$^{-1}$ peak [44]. The band associated with Ti-O-Ti lattice vibration at 1384 cm$^{-1}$ can be observed [45]. The broad band centered at 480 cm$^{-1}$ and 548 cm$^{-1}$ could be associated with the Ti-O-C stretching and vibrations, respectively [32]. In the spectrum of MWCNT-Ag$^+$/Ti$^{4+}$ = 0.5%, more bands can be identified and associated with the presence of the COOH (peak from 1744 cm$^{-1}$) or C-O-C vibrations (peak from 1084 cm$^{-1}$) [46,47]. They can also be seen in less intensity in the other FTIR spectra of MWCNT–TiO$_2$–Ag samples, suggesting the presence of the MWCNT structures in the composition of all these hybrid materials.

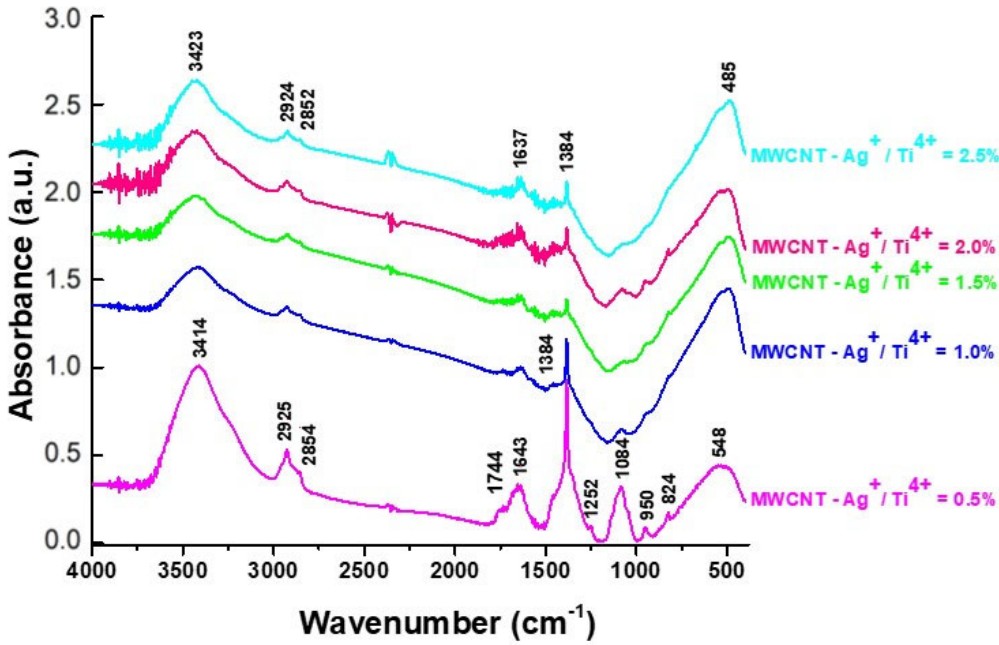

**Figure 4.** The FT–IR spectra of MWCNT–TiO$_2$–Ag.

### 2.4. Raman Spectroscopy

The typical Raman active modes of anatase TiO$_2$ are composed of four main peaks around 152 (E$_g$), 389 (B$_{1g}$), 510 (A$_1$), and 633 cm$^{-1}$ (E$_g$). In addition to these bands, a small peak, assigned to the E$_g$ mode, is localized at about 189 cm$^{-1}$ [48]. The MWCNT–TiO$_2$–Ag nanocomposites contain supplementary bands characteristic of the carbon nanostructures, Figure 5. Thus, a D band appeared at about 1361 cm$^{-1}$, being specific to the presence of defects in the carbon system, while the G peak from about 1589 cm$^{-1}$ is related to the vibration of the sp$^2$—bonded carbon atoms [49]. A 2D band (also called 'G') is present at about 2712 cm$^{-1}$ in the spectrum of the MWCNT-Ag$^+$/Ti$^{4+}$ = 2.5% sample and 2718 cm$^{-1}$ in the CNT precursor sample.

### 2.5. Electron Spin Resonance Measurements

EPR spectra are shown in the Figure 6a,b. The analysis of EPR spectra reveals a narrow resonance signal centered at g = 2.004 due to an oxygen vacancy with a trapped electron named F-center [50]. This signal diminishes by increasing the Ag content in the TiO$_2$ lattice. Besides this signal, at the higher and lower field side, two low resonance signals are centered at g = 1.97 and g = 2.03, as shown in the inset of Figure 6a. The signal at g = 1.97 was attributed to Ti$^{3+}$ ions in the anatase crystalline phase, and the signal at g = 2.03 showed the presence of surface O$^-$ ions [50].

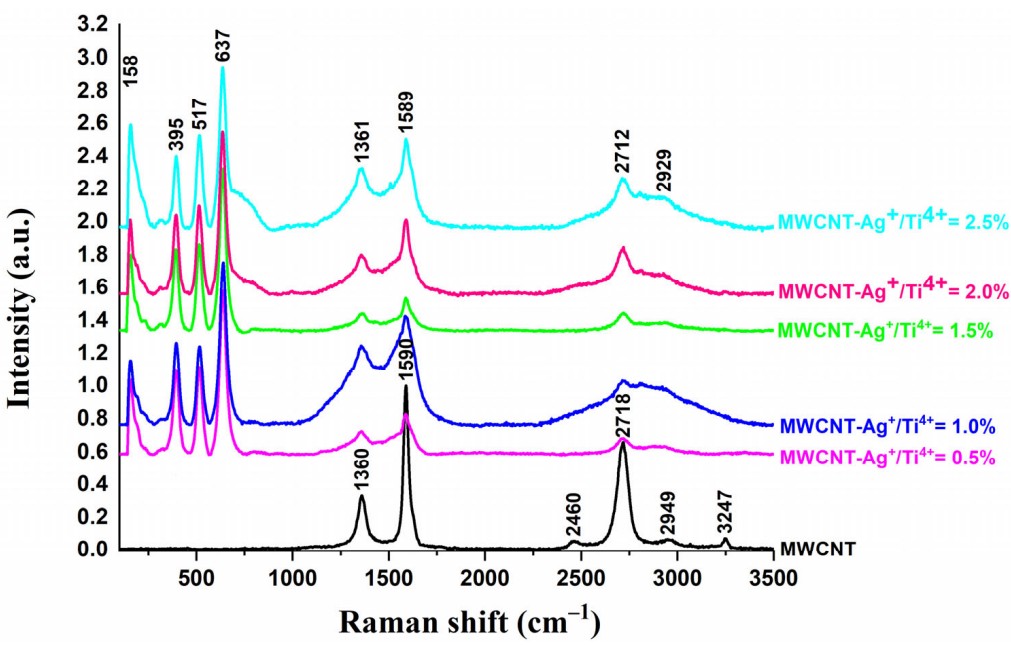

**Figure 5.** Raman spectra of MWCNTs-TiO$_2$ with different amounts of Ag nanoparticles were recorded with a 514 nm excitation laser line.

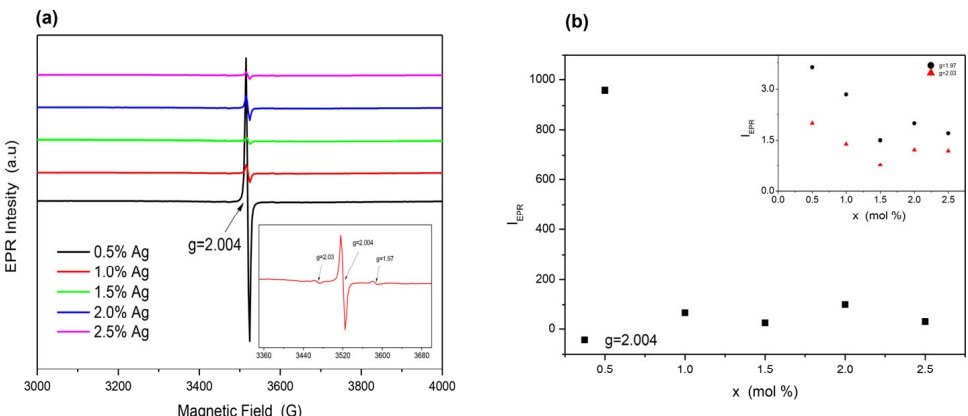

**Figure 6.** (**a**) EPR spectra of MWCNT–TiO$_2$–Ag and (**b**) EPR integral intensity of the resonance signal for the analyzed samples.

The EPR integral intensity of the resonance signal, which is proportional to the number of electrons spin from the sample, is illustrated in Figure 6b. It could be observed that all resonance signals show the same behavior, decreasing with the increase of dopant concentration up to 1.5% mol Ag, followed by a slight increase. The oxygen vacancies concentration decreases in samples with high Ag content (≥1.5 mol%).

### 2.6. UV–ViS Spectroscopy

Figure 7 indicates the UV–VIS spectra of MWCNTs-TiO$_2$ with different Ag nanoparticle amounts. An intense absorption is observed at 353 nm, specific to the intrinsic band gap of TiO$_2$ nanoparticles [51]. The broad absorption, that covers the visible range is caused by the addition of MWCNTs [15]. Moreover, the strong absorption peak around 470–580 nm arises from the surface plasmon absorption due to Ag particles [52]. By increasing the Ag concentration, a red shift to a higher wavelength is observed in the absorption edge of the nanocomposites due to electronic interaction between MWCNT and TiO$_2$-modified Ag [53]. Supplementary, the spectrum contains a very low absorption peak at 263 nm, attributed to the transition to Ag's higher energy state of valence electrons [54].

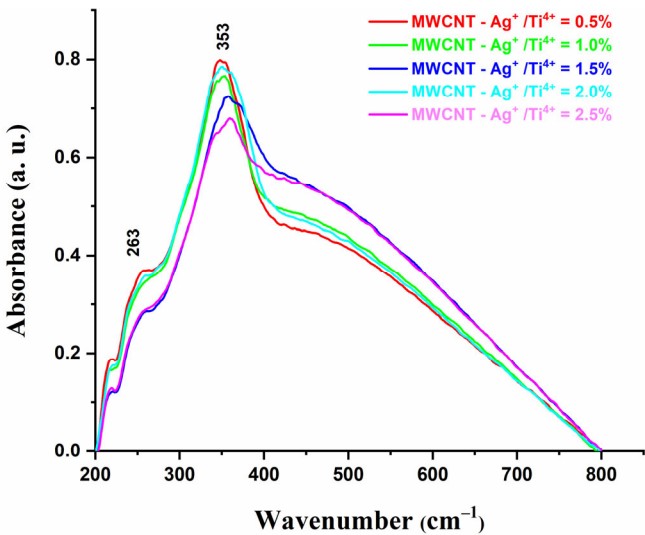

**Figure 7.** UV–VIS spectrum of MWCNTs–TiO$_2$–Ag.

### 2.7. Optical Band Gap Energy, E$_g$ Determination

The influence of MWCNT and Ag on the TiO$_2$ nanoparticle's band gap energy (E$_g$) was estimated using a Tauc plot [55,56]. The band gap values for direct transitions are 2.41–2.77 eV (Figure 8) (Table 2). The spectra redshift was observed similar to that reported by Li et al. [57].

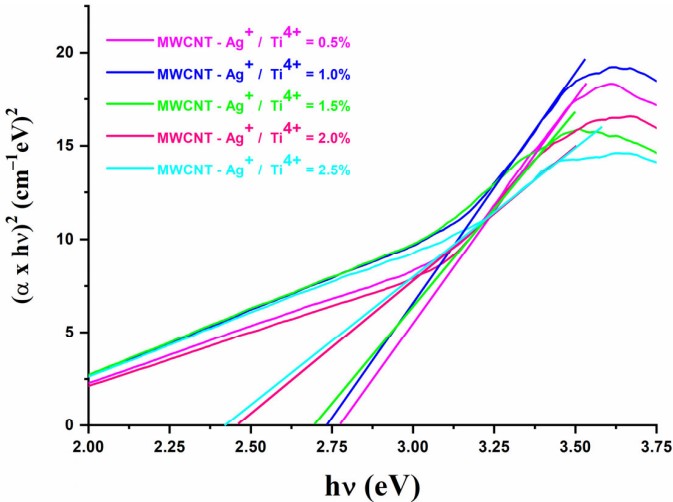

**Figure 8.** Extrapolation of the band gap energy for direct transition.

**Table 2.** Optical band gap energy of TiO$_2$ nanoparticles calculated based on UV–VIS absorption spectra of MWCNT–TiO$_2$–Ag nanocomposites.

| Sample | E$_g$—Direct Transition |
|:---:|:---:|
| MWCNT-Ag$^+$/Ti$^{4+}$ = 0.5% | 2.77 |
| MWCNT-Ag$^+$/Ti$^{4+}$ = 1.0% | 2.73 |
| MWCNT-Ag$^+$/Ti$^{4+}$ = 1.5% | 2.69 |
| MWCNT-Ag$^+$/Ti$^{4+}$ = 2.0% | 2.46 |
| MWCNT-Ag$^+$/Ti$^{4+}$ = 2.5% | 2.41 |

### 2.8. Morphology of Nanocomposites

Conventional TEM images (Figure 9a,b) revealed the presence of polyhedral nanoparticles over the nanotube surface, tending to agglomerate.

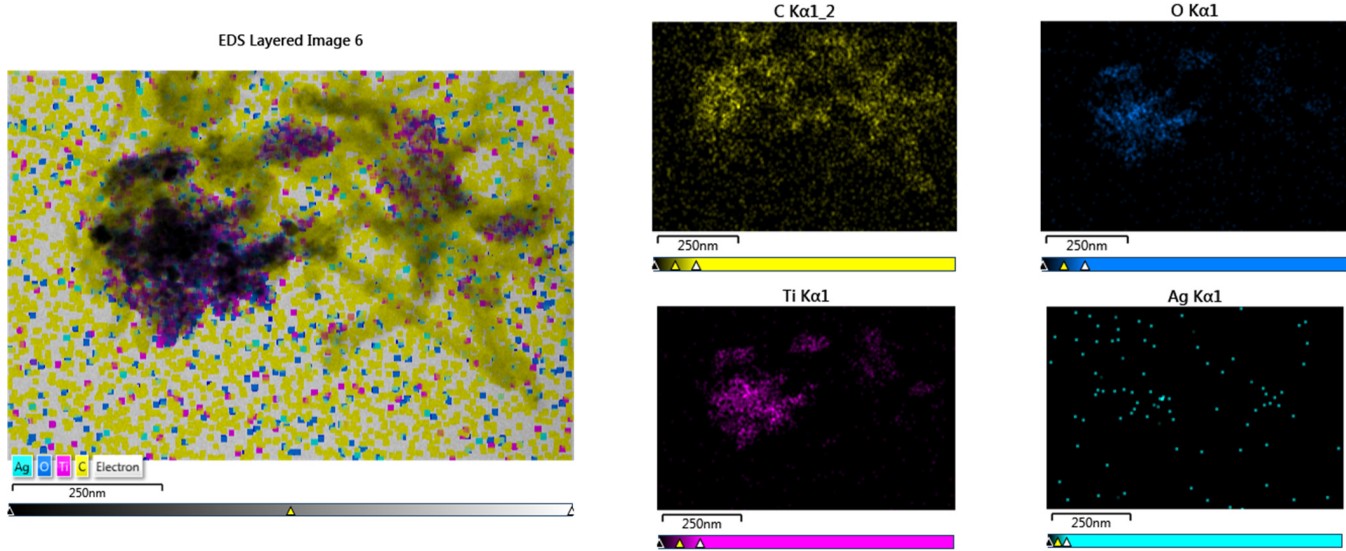

**Figure 9.** TEM images of MWCNT–TiO$_2$–Ag. (**a**) Ag$^+$/Ti$^{4+}$ = 0.5% and (**b**) Ag$^+$/Ti$^{4+}$ = 2.5%.

In addition, the EDX shown in Figure 10 reveals the presence of elements including C, Ti, O, and Ag in the MWCNT–TiO$_2$–Ag samples.

**Figure 10.** EDX mapping of MWCNTs-Ag$^+$/Ti$^{4+}$ = 1.5%.

### 2.9. Photoluminescence Spectroscopy

The PL spectra of MWCNTs-TiO$_2$ with different amount of Ag nanoparticles is illustrated in Figure 11. As can be seen in the PL spectra, the intensity of the PL bands decreases with the incorporation and increase of the content of Ag$^+$ ions in the TiO$_2$ lattice because Ag nanoparticles act as traps to capture the photo-reduced electrons and thus inhibit the recombination of the electron–hole pair. The peak at ~470 nm is due to defect states formed by oxygen vacancies [58], and the peak from 330 nm is associated with the band—the gap of titania [59]. Emissions around ~550 nm are associated with electron transition from $[TiO_6]$ to O on the surface [60].

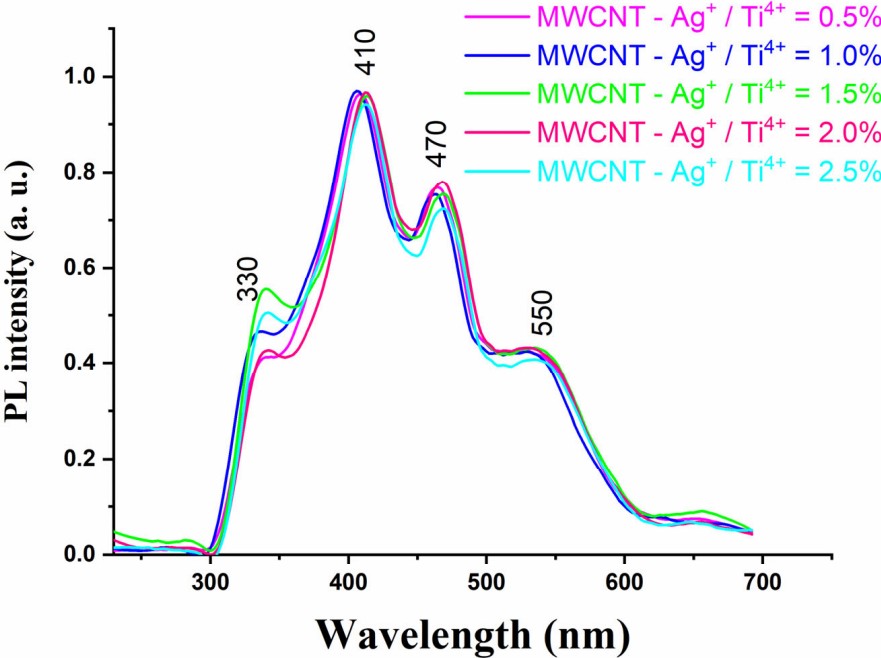

**Figure 11.** PL of MWCNT–TiO$_2$–Ag nanocomposites.

### 2.10. Photocatalytic Activity

The photocatalytic activity of the samples was investigated using a synthetic solution of Allura Red-E129 under UV irradiation. Since the degradation process's efficiency depends on the concentration of the pollutants and the amount of the photocatalyst in the studied systems, minimal amounts of each component were chosen for the study. Before irradiation, the samples were kept in the dark for 60 min to achieve adsorption–desorption equilibrium. The testing of the photocatalytic activity of MWCNT decorated with TiO$_2$–Ag nanoparticles was carried out by tracking the spectrophotometric decrease over time of the concentration of pigment E129, $\left(c_0 = 2 \cdot 10^{-5}\right)$ ($\lambda$ = 502 nm), following the adsorption and photodecomposition of nanostructures. Figure 12 shows the degradation efficiency of the samples. The adsorption capacity of the samples increases with Ag content.

The efficiency of the photocatalysts for selected Allura Red degradation was calculated according to Equation (1):

$$F(\%) = \frac{C_0 - C_t}{C_0} \times 100 \tag{1}$$

where Co is the initial dye concentration and Ct is the concentration of colorant at time t. The nanostructures with 2.0% Ag have the best photocatalytic activity for the degradation of Allura Red under UV irradiation, demonstrating a degradation efficiency of 79.99% (Figure 12). According to literature data, the photocatalytic activity of Ag-doped titania is dependent on the concentration of noble metal particles [61,62], and the diminishing of the photoreaction at higher loading is due to the so-called "screening effect" [63]. The presence of Ag nanoparticles stimulates the separation of charges, but too high a concentration

of Ag particles shields the active centers of the semiconductor, and thus reducing its photocatalytic activity.

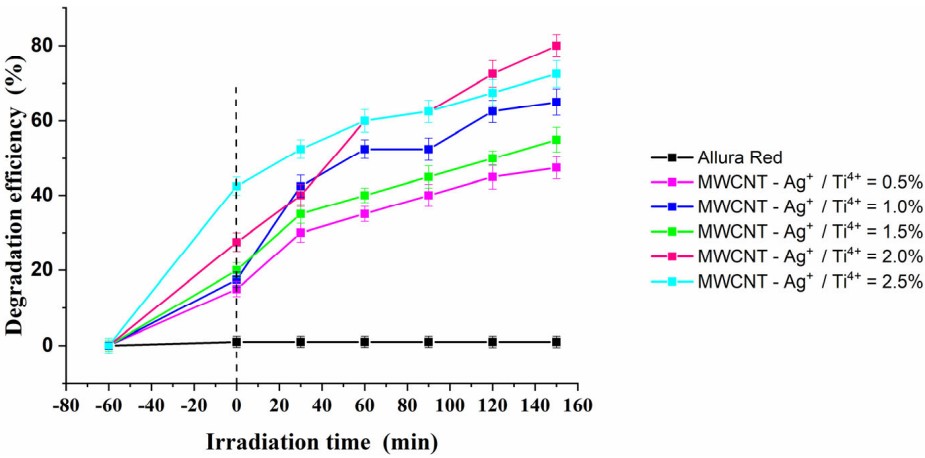

**Figure 12.** Photocatalytic degradation of Allura Red by MWCNT nanostructures The error bars indicate standard deviation (n = 3).

According to the Langmuir–Hinselwood expression: $-\ln\left(\frac{C_t}{C_o}\right) = kt$, under $C_0$ and $C_t$ are the dye concentration at time 0 and t (after irradiation), respectively, k (min$^{-1}$) is the pseudo first-order rate constant, and the best photocatalyst is the one with 2.0% Ag (Figure 13), as mentioned above. The obtained results are shown in Table 3.

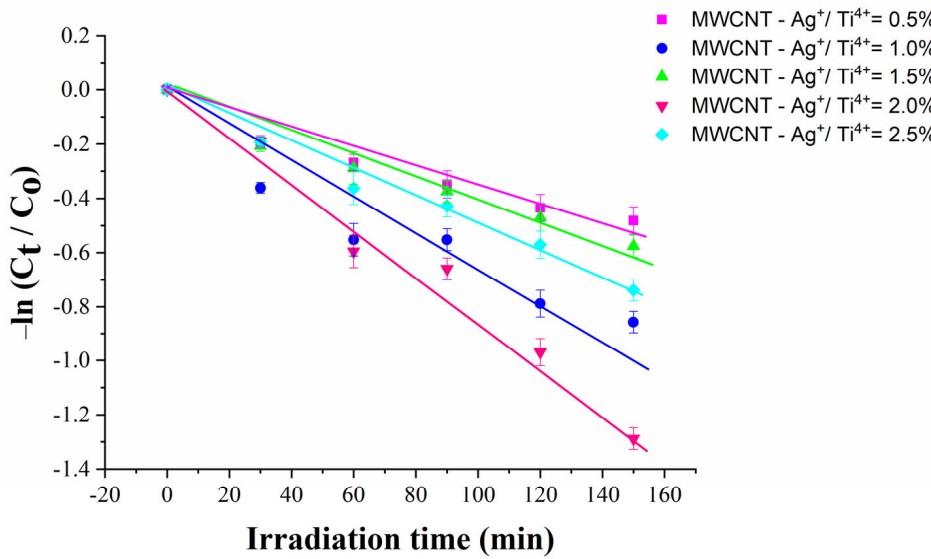

**Figure 13.** Kinetics of Allura Red degradation by nanostructures. The error bars indicate the standard deviation (n = 3).

**Table 3.** Photodegradation rate, apparent first-order rate constant ($k_i$) of photocatalytic degradation, and correlation coefficient ($R^2$).

| Sample | Photodegradation Rate (%) | $k_i \cdot 10^{-3}$ | $R^2$ |
|---|---|---|---|
| MWCNT-Ag$^+$/Ti$^{4+}$ = 0.5% | 47.5 | 3.06 | 0.95097 |
| MWCNT-Ag$^+$/Ti$^{4+}$ = 1.0% | 65 | 5.21 | 0.95263 |
| MWCNT-Ag$^+$/Ti$^{4+}$ = 1.5% | 55 | 3.57 | 0.97206 |
| MWCNT-Ag$^+$/Ti$^{4+}$ = 2.0% | 80 | 8.42 | 0.98006 |
| MWCNT-Ag$^+$/Ti$^{4+}$ = 2.5% | 72.5 | 4.66 | 0.98538 |

### 2.10.1. Scavanger Experiments

To evaluate the presence and relevance of ROS, organic and inorganic substances, namely scavengers that can interfere with ROS formation, have been applied to different advanced oxidative processes. To scavenge the hole in the valence band, electron donors such as oxalate and formic acid are employed [64–67], and alcohols, such as 2-propanol, t-butanol, and methanol, are usually applied to confirm the contribution of HO· [64,67,68].

To determine the role of electron-hole scavengers on the photodegradation performance of MWCNT-Ag$^+$/T$^{4+}$ = 2.0%, the experiment was carried out by analyzing the degradation activities of Allutra Red solution with an initial concentration of $2 \times 10^{-5}$. Figure 14 shows the addition of vitamin C for $O_2^{·-}$ and tert butanol for $OH^·$, act as scavengers exhibit photodegradation efficiency within 150 min. The percentage of Allura Red degradation after 150 min in the presence of vitamin C was 6.74%, and for tert-butanol was 8.1% (Figure 14), demonstrating the relevance of hydroxyl and singlet oxygen radicals to Allura Red.

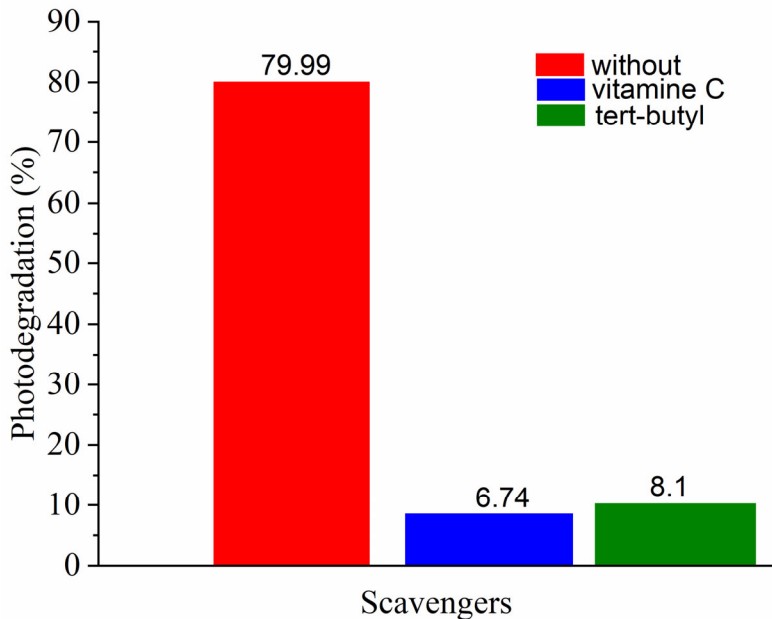

**Figure 14.** Photocatalytic degradation of Allura Red solution containing scavengers.

### 2.10.2. Effect of the Solution pH on Pollutant Eliminations

The effect of the initial solution pH in the range of 2–10 on the degradation of Allura Red (Figure 15) revealed that pH = 6 provided the highest photodegradation efficiency. A drastic decrease of the photocatalytic performance at pH = 10 is probably due to the repulsion between the photocatalyst and pollutant molecules, both with negative charge [69]. The decrease in the photocatalyst performance at pH = 3 is mainly assigned to the dissolution of the photocatalyst [70].

### 2.10.3. Photocatalyst Cycling Experiments

The recyclability of photocatalysts is essential for their practical applications. After each use, the photocatalyst is centrifuged and washed several times with deionized water and dried overnight. The photodegradation rate remains almost constant after 150 min, after three cycles (Figure 16) of use, without any treatment or activation, indicating good stability of the catalyst. The gradual decrease in photocatalytic performance is probably related to the partial loss of nanoparticles by centrifugal washing during the cycling of the material.

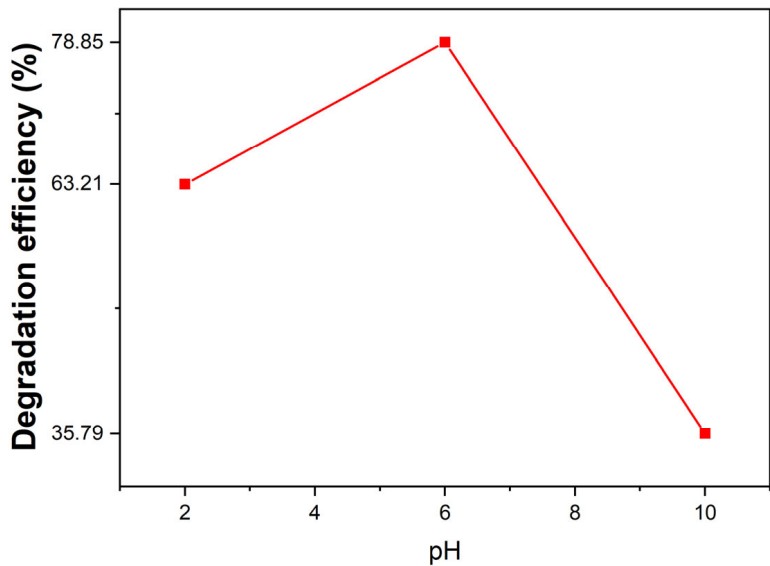

**Figure 15.** Effect of initial solution pH on Allura Red dye.

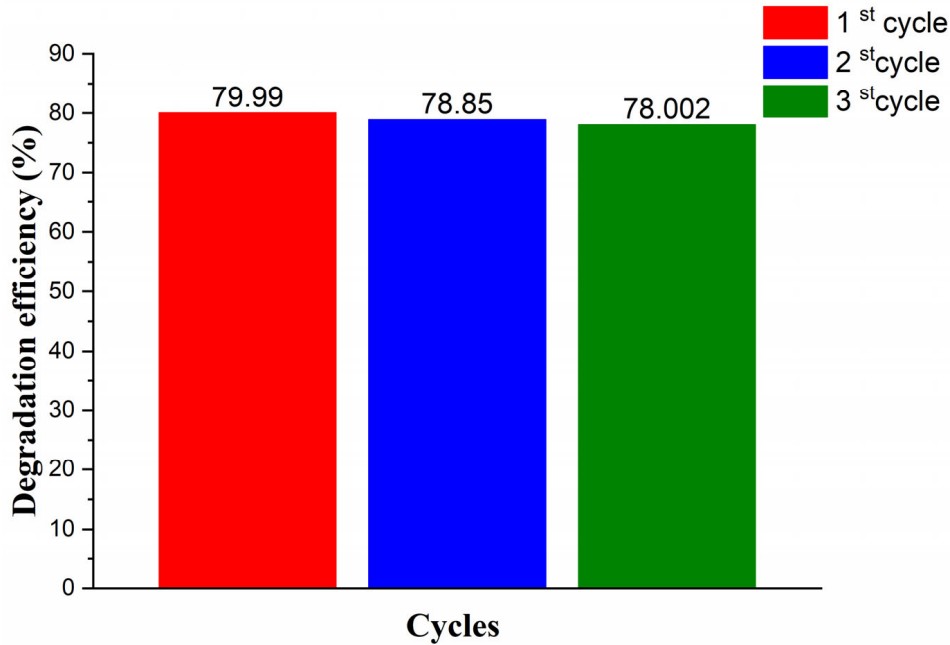

**Figure 16.** The reusability of MWCNT-Ag$^+$-Ti$^{4+}$ for degradation of Allura Red.

2.10.4. Photocatalytic Mechanism

Reactiv Oxigen Species (ROS) Generation

ESR coupled with the spin-trapping method was used to identify the ROS involved in the photocatalytic process of MWCNT–TiO2–Ag nanocomposites. As a spin-trapping agent, we used DMPO. The experimental spectrum obtained after 30 min UV irradiation of DMPO-MWCNT–TiO$_2$–Ag solution is shown in Figure 17. The obtained spectrum is complex, consisting of many resonance lines due to different spin adducts generated at the time of irradiation. The spectrum simulation was performed to identify and extract the contribution of the spin adducts. The simulated spectrum represents the linear contribution of the following spin adducts: ●DMPO-OCH$_3$ (=13.2 G, =8.3 G, =2.0 G, relative concentration 23.8%), ●DMPO-O$_2^-$ (=13.0 G, =11.3 G, =2.6 G, relative concentration 37.9%), ●DMPO-OOH (=13.5 G, =11.0 G, =0.6 G, relative concentration 17.2%), nitroxide-like radical (=13.9 G, relative concentration 21%). The presence of ●DMPO-OCH$_3$ spin adducts is proof of the

presence of ●OH radicals due to the interaction between the DMSO solvent and ●OH radicals [71]. ●OOH radical is obtained by the protonation of superoxide radical ●$O_2^-$ [61]. Nitroxide-like radical appears by cleavage of the N–C bond and ring opening of DMPO [22]. These results attest the presence of both ●OH and ●$O_2^-$ radicals. The relative concentration of ●$O_2^-$ radicals has been more than twice that of the hydroxyl radical concentration.

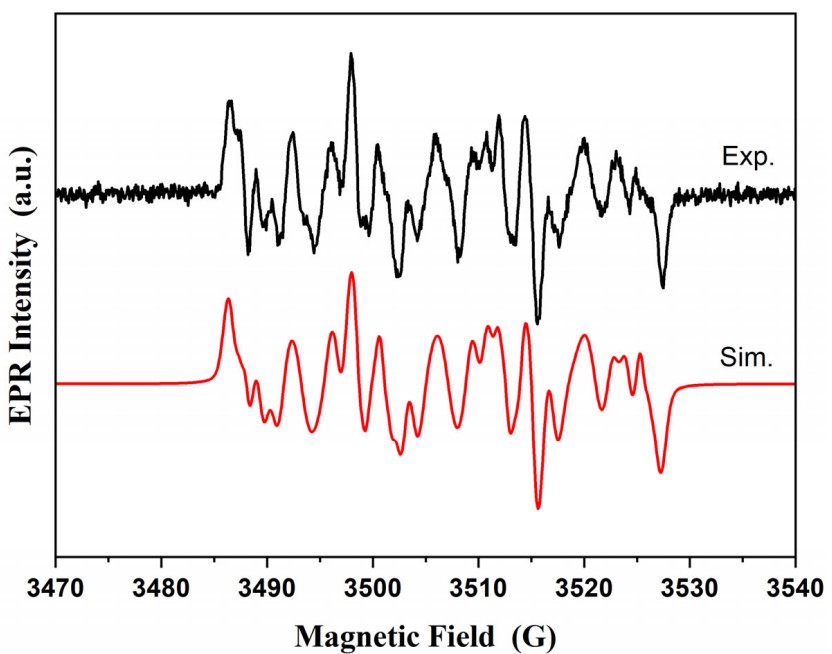

**Figure 17.** Experimental and spectrum simulation of DMPO spin adducts generated by MWCNT–$TiO_2$–Ag sample after 30 min of irradiation.

To explain the charge carriers behavior and the role of defects, the photocatalytic mechanism was proposed together with the energy band diagram for MWCNT–$TiO_2$–Ag nanocomposites. Under UV light irradiation, the electrons from the $TiO_2$ valence band (VB) are excited into the conduction band (CB), leaving an equal number of holes ($h^+$) in the VB, or indirectly can be transferred to the CB through the defects' sub-band (Figure 18). EPR measurements evidenced the presence of $Ti^{3+}$ defects and oxygen vacancies. The excited electrons from the $TiO_2$ CB can be transferred to Ag nanoparticles due to the higher work function of Ag [72]. Consequently, a better separation of the photogenerated charges was ensured, and the electrons from Ag nanoparticles and MWCNT can react with $O_2$ molecules and generate superoxide radicals ($O_2^-$). In addition, the $h^+$ from the $TiO_2$ VB will pass to the MWCNT via impurity levels in the bandgap [73] and, together with $h^+$ from Ag, interact with the $H_2O$ molecules, generating ●OH reactive species. These reactive species are responsible for the degradation of dye molecules.

Analyzing the EPR and photocatalysis results shows that a medium concentration of defects observed in the case of the sample with $Ag^+/Ti^{4+}$ = 2% is beneficial for dye degradation. Similar results were obtained in the case of $WO_3$ thin films, where there was evidence that in the case of samples with low oxygen vacancies concentration, a reduced charge separation was observed [74]. When the oxygen vacancies exceed an optimum concentration, the defects act as trap charges, increasing the recombination pathways and decreasing the photocatalytic performance.

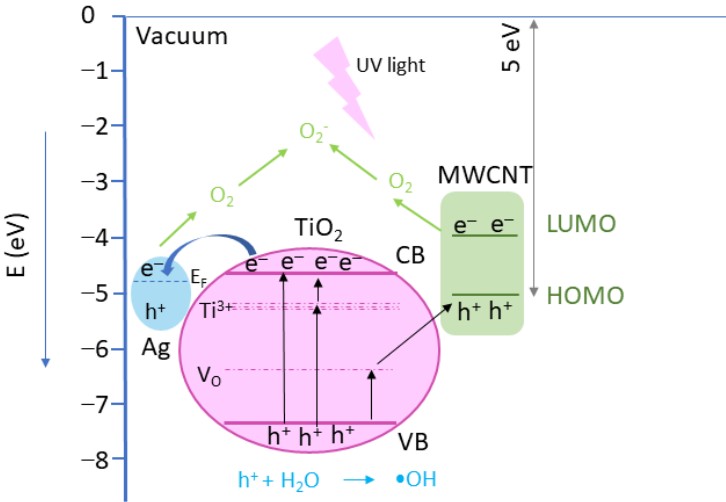

**Figure 18.** The MWCNT—TiO$_2$—Ag samples' energy band diagram shows the proposed photocatalytic mechanism. The energy level positions were drawn according to [74,75].

## 3. Materials and Methods

### 3.1. Materials

Materials and reagents used for the preparation of MWCNT-TiO$_2$:AgX% were: multi-walled carbon nanotubes (MWCNTs) with a 99% purity, purchased commercially (Sigma-Aldrich, Merck, KGaA, Darmstadt, Germany), poly-allylamine hydrochloride (PAH) (Alfa-Aesar, Thermo Fisher (Kandel) GmbH, Erlenbachweg, Germany), sodium chloride, NaCl-(Alpha Aesar, Bio Aqua Group, Targu Mures, Romania), titanium tetraisopropoxide Ti[OCH(CH$_3$)$_2$]$_4$, note Ti(Opri)$_4$ (Fluka Chemie, GmBH, Sigma-Aldrich Chemie, Steinheim, UK), acetylacetone, C$_5$H$_8$O, note AcAc (Merck, KgaA, Darmstadt, Germany), AgNO$_3$ (VWR Chemicals, UK), L+ ascorbic acid (vitamin C), (Merck, KgaA, Darmstadt, Germany), tert-butyl alcohol (Merck, KgaA, Darmstadt, Germany) absolute ethanol (C$_2$H$_5$OH-EtOH) (Alpha Aesar, Bio Aqua Group, Targu Mures, Romania). All chemicals were of analytical grade and used without further purification. The aqueous solutions were prepared with Milli-Q water from a Direct-Q 3UV system (Millipore, Bedford, MA, USA).

### 3.2. Sample Preparation

To obtain carbon nanotubes (MWCNT) decorated with TiO$_2$–Ag (MWCNT–TiO$_2$–Ag), several synthesis steps were performed, as follows: in the first stage MWCNT were functionalized with -OH and -COOH groups, and in the second stage the TiO$_2$:AgX% nanoparticles previously prepared by modified Pechini method were attached on functionalized MWCNT through polyallylamine hydrochloride (PAH).

#### 3.2.1. Functionalization of MWCNTs

Functionalization of MWCNTs was performed by acid treatment of commercial MWCNTs (D × L 110–170 nm × 5–9 μm, Aldrich) in 200 mL mixture of H$_2$SO$_4$:HNO$_3$ (3:1 vol ratio) in an ultrasound bath for 4 h. Then, the content was cooled, centrifuged, and washed with distilled water several times to maintain its neutralization. Further, it was dried at 65 °C in the oven to obtain its functionalized MWCNTs.

#### 3.2.2. Synthesis of TiO$_2$ Modified by Ag

To obtain TiO$_2$ modified by Ag nanopowder by Pechini method, titanium tetraisopropoxide Ti[OCH(CH$_3$)$_2$]$_4$, and acetylacetone, C$_5$H$_8$O, is used as precursor. The mixture, as obtained, a different Ag$^+$/Ti$^{4+}$ atomic ratio, respectively 0.5, 1.0, 1.5, 2.0, and 2.5% of AgNO$_3$ (note as Ag$^+$/Ti$^{4+}$ = 0.5%, Ag$^+$/Ti$^{4+}$ = 1.0%, Ag$^+$/Ti$^{4+}$ = 1.5%, Ag$^+$/Ti$^{4+}$ = 2.0% and Ag$^+$/Ti$^{4+}$ = 2.5%) is added in solution, after 2 h. The L(+)-ascorbic acid was added by

continuous stirring. The $AgNO_3$:L(+)-ascorbic acid was 1:1 (%wt). This mixture was dried at room temperature for 10 days and then calcinated, in air, at 455 °C, for 4 h.

### 3.3. Decoration of MWCNT with TiO$_2$ Modified by Ag

To ensure good attaching, the functionalized MWCNTs were modified by polymer wrapping with poly (allylamine hydrochloride) (PAH). The nanotubes (8 mg) were dispersed in a 0.5 wt% PAH salt solution (0.5 M NaCl, 500 mL) and sonicated for 4 h, then stirred overnight at 80 °C. The excess polymer was removed by repeated centrifugation and redispersed in water until a stable, homogenous MWCNT suspension was obtained. The amine functionalities on the MWCNTs surface (MWNT-PAH) ensure good separation and stability due to electrostatic interactions (repulsions) in an aqueous solution. Before the decoration process, the functionalized MWCNT and $TiO_2$:Ag nanoparticles were dispersed separately by sonication in ethanol for 1 h and then were mixed. The process is continued for another 4 h. The obtained MWCNT decorated with $TiO_2$:Ag (MWCNT-$TiO_2$-Ag) were separated by centrifugation, washed with distilled water several times, and dried in an oven at 70 °C.

### 3.4. Characterization

Thermal analysis (TG-DTA-DTG) was recorded with a Mettler–Toledo Thermogravimeter 851e (Columbus, OH, USA) equipment. The TG-DTA-DTG was performed in air, in the 20–1000 °C temperature range using upgraded computer-controlled equipment. About 38,679 mg of sample was heated in Pt-holder with another Pt-holder containing a-alumina as reference material. The sample was heated at 10 °C/min from ambient temperature to 1000 °C in static air.

The XRD was recorded on the BRUKER D8 Advance X-ray diffractometer (Rheinstetten, Baden-Württemberg, Germany), working at 45 kV and 45 mA. The $Cu_{K\alpha}$ radiation, Ni filtered, was collimated with Soller slits. A germanium monochromator was used. The data of the X-ray diffraction patterns were collected in a step-scanning mode with steps of $\Delta 2\theta = 0.01°$. Pure silicon powder (standard sample) was used to correct the data for instrumental broadening. Crystallographic identification was accomplished by comparing the experimental XRD patterns with MATCH software (Kreuzherrenstr, Germany) version 1.11.

The Warren–Averbach X-ray profile Fourier analysis of the (101), (004), (200), and (204) anatase peak profiles were processed by the XRLINE [38] computer program to determine the effective crystallite mean size ($D_{eff}$). The crystallite size distribution function was determined from the second derivative of the strain-corrected Fourier coefficients [41].

FT–IR spectra of the powder samples using the KBr pellet technique, in the absorbance mode have been recorded using JASCO FT/IR-6100 Fourier Transform Infrared Spectrometer (JASCO International Co., Ltd., Tokyo, Japan) in the 400–4000 $cm^{-1}$ wavenumber range with a resolution of 4 $cm^{-1}$.

The Raman spectra have been recorded at room temperature with a JASCO NRS 3300 spectrophotometer (JASCO International Co., Ltd., Tokyo, Japan) equipped with a CCD detector (−69 °C) using the laser excitation wavelength of 515 nm line of an Ar-ion laser was used as the excitation source and a laser power of 0.7 mW. A 100× Olympus objective, an exposure time of 60 s, and three accumulations were used for each spectral measurement. The spectrometer was calibrated using the Si Raman peak from 521 $cm^{-1}$.

EPR measurements of powder samples were carried out on a Bruker E-500 ELEXSYS X-band (9.52 GHz) spectrometer (Rheinstetten, Baden-Württemberg, Germany) at room temperature under identical conditions: microwave frequency of 9.5248 GHz, microwave power 2 mW, the modulation frequency of 100 kHz and modulation amplitude 10 G.

For the morphology of samples, a Scanning Electron Microscope Hitachi SU8230 scanning electronic microscope (Tokyo, Japan), using 30 kV, 15 mm working distance. The instrumental analysis of the sample composition was determined by Oxford Instruments EDS System (Oxford, UK) and AZtech software (Greeley, CO, USA).

Optical UV–VIS absorption spectra were recorded on the PERKIN-ELMER LAMBDA 45 spectrophotometer (JASCO International Co., Ltd., Tokyo, Japan), equipped with an integrating sphere assembly of 200–900 nm.

The fluorescence spectra were obtained using an ABL&JASCO V 6500 spectrofluorometer (Tokyo, Japan) with a xenon lamp of 150 W.

Photocatalytic measurements were performed by immersing the 0.5 mg samples in the Allura Red solution $\left(c_0 = 2 \times 10^{-5}\right)$ irradiated with UV lamp (15 W) emitting at 365 nm. The absorbance of the Allura Red solution was measured using a T80+ UV–VIS, Pro Instruments Ltd. Spectrophotometer (Leicestershire, UK). The effect of the initial 2–10 pH solution on the degradation of Allura Red, under a UV lamp (15 W) emitting at 365 nm for 150 min was also determined.

ESR coupled with the spin-trapping probe technique was employed. 5,5-dimethyl-1-pyrroline N-oxide (DMPO, Sigma-Aldrich, Merck, KGaA, Darmstadt, Germany) was used as a spin-trapping reagent. The nanoparticles (10 mg) were dispersed in DMSO (1 mL) and homogenized in an ultrasound bath (30 min) before use. DMPO of 0.2 mol/L concentration was added to the suspension. The samples were prepared immediately before measurements and transferred into the quartz flat cell optimized for liquid measurements.

## 4. Conclusions

The present paper reported the synthesis and characterization of MWCNT–$TiO_2$–Ag nanocomposites for photocatalytic oxidation of Allura Red. The investigation results show that samples have crystalline structures containing anatase phases with an average crystallite size of 14 nm. The nanocomposites have a polyhedral shape and are distributed over the nanotube surface with a tendency to agglomerate, as evidenced by TEM. FT–IR spectra suggest the presence of the vibrations corresponding to MWCNT–$TiO_2$–Ag structures in the nanocomposites. The existence of active modes of anatase $TiO_2$ and vibration of the $sp^2$-bonded carbon atoms characteristic of graphite are evidenced by Raman characterization. The presence of oxygen vacancies was evidenced by EPR spectroscopy, and their concentration decreased with the increasing dopant concentration up to 1.5% mol, followed by a slight increase in accordance with the PL results.

Moreover, a narrowing of the band gap by doping was observed. The highest photocatalytic activity (79.99%) was achieved for the sample containing 2% Ag. A higher concentration of Ag particles shields the active centers of the semiconductor, thus reducing its catalytic activity. Moreover, the generation under UV light irradiation of both active radicals OH• and •$O_2^-$ have been highlighted by the ESR spin-trapping technique. The •$O_2^-$ are the main species generated and responsible for Allura Red degradation. The mechanism of photocatalytic activity was explained based on the ROS generated, scavenger experiments, and band energy diagram.

**Author Contributions:** Conceptualization, M.S. and R.-C.S.; investigation, M.S., A.P., D.T., C.B.-G., M.Z. and C.T.; methodology, M.S. and R.-C.S.; writing—original draft, M.S. and R.-C.S.; writing—review and editing, M.S., R.-C.S., A.P. and D.T. All authors have read and agreed to the published version of the manuscript.

**Funding:** This research was funded by the MCID through the "Nucleu" Program within the National Plan for Research, Development, and Innovation 2022–2027, project PN 23 24 01 03 and by a grant of the Romanian Ministry of Education and Research. CNCS-UEFISCDI, project PN-III-P1-1.1-TE-2021-0661 within PNCDI III. Also, the work was supported by Program 1—Development of the national research and development system, Subprogram 1.2—Institutional performance—Projects that finance the RDI excellence, Contracts no. 37PFE/30.12.2021.

**Data Availability Statement:** The raw/processed data required to reproduce these findings cannot be shared at this time as the data form part of an ongoing study. Part of the research data required to reproduce these findings can be shared, however, upon appropriate request.

**Conflicts of Interest:** The authors declare no conflict of interest.

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
