# Peer review of "The Influence of Ag+/Ti4+ Ratio on Structural, Optical and Photocatalytic Properties of MWCNT–TiO2–Ag Nanocomposites"

_inorganics, doi:10.3390/inorganics11060249_

Round 1
Reviewer 1 Report
The presented study is of particulate interest and has multiple characterization analyses. However I suggested the following comments for further improvement:
1- Some grammatical / mistyping errors were detected, English editing / revision should be provided.
2- Line 46 (introduction), needs more elaboration with a proper reference.
3- Introduction part should be improved by making a comparison of previously reported work, as the novelty part is not clear.
4- Why authors choose Allure Red (E19), please justify.
5- Photocatalytic part should be improved by adding reaction mechanism with energy levels, effect of pH, stability of performance test (reusability), etc...
1- Some grammatical / mistyping errors were detected, English editing / revision should be provided.
Author Response
Dear Reviewer,
Please find in the attached file the response to your comments.

Reviewer 2 Report
This present work shows the fabrication of composites as photocatalyst containing carbon nanotubes with TiO2-Ag for dye degradation. The current version is overall good and some modificiation and experiments should be further performed before acceptance.
1. Why high Ag content helped to reduce the oxygen vacancy? How the oxygen vacancy affects photocatalytic performance?
2. It is unreasonable to directly determine bandgap energy for composites (Figure 8).
3. The authors should explain why different composites showed almost same PL intensity as it is a essential technology to characterize separation of electrons and holes.
4. The correlation coefficiency is not satisfied. The photocatalytic test should be repeated to show the error bar.
5. Scavenger tests and cycling test should be supplemented.
Overall is good
Author Response
Dear reviewer,
Please find the in the attached file the response to your comments.

Round 2
Reviewer 1 Report
Authors addressed raised comments properly. The presence article is acceptable for publication.
Reviewer 2 Report
The present version could be accepted